# POINTER-CAD: UNIFYING B-REP AND COMMAND SEQUENCES VIA POINTER-BASED EDGES & FACES SELECTION

## ABSTRACT

Constructing computer-aided design (CAD) models is labor-intensive but essential for engineering and manufacturing. Recent advances in Large Language Model (LLM) have inspired the LLM-based CAD generation by representing CAD as command sequences. But these methods struggle in practical scenarios because command sequence representation does not support entity selection (e.g. faces or edges), limiting its ability to support complex editing operations such as *chamfer* or *fillet*. Further, the discretization of a continuous variable during *sketch* and *extrude* operations may result in topological errors. To address these limitations, we present **Pointer-CAD**, a novel LLM-based CAD generation framework that leverages a pointer-based command sequence representation to explicitly incorporate the geometric information of B-rep models into sequential modeling. In particular, Pointer-CAD decomposes CAD model generation into steps, conditioning the generation of each subsequent step on both the textual description and the B-rep generated from previous steps. Whenever an operation requires the selection of a specific geometric entity, the LLM predicts a *Pointer* that selects the most feature-consistent candidate from the available set. Such a selection operation also reduces the quantization error in the command sequence-based representation. To support the training of Pointer-CAD, we develop a data annotation pipeline that produces expert-level natural language descriptions and apply it to build a dataset of approximately **575K CAD models**. Extensive experimental results demonstrate that Pointer-CAD effectively supports the generation of complex geometric structures and reduces segmentation error to the order of $10^{-3}$, **a $100\times$ improvement** over prior methods, thereby significantly mitigating the topological inaccuracies introduced by quantization error.

## 1 INTRODUCTION

Computer-Aided Design (CAD) plays an essential role in modern engineering, enabling precise and efficient design across diverse industry domains Rapp et al. (2021); Castellino (2005). The conventional CAD design workflow typically begins with 2D sketches(e.g. *lines, circles*), progresses to 3D modeling operations(e.g. *extrude, chamfer, fillet*), and culminates in models stored in Boundary Representation (B-rep) (Lambourne et al., 2021) format by software. However, this process remains heavily reliant on manual input, making it time-consuming, particularly for intricate designs.

Recent efforts (Wu et al., 2025a; Xu et al., 2024b; Alam & Ahmed, 2024; Xu et al., 2022) in CAD generation have explored parametric design synthesis with large generative models, aiming for fully autonomous CAD creation in an autoregressive manner. Inspired by the reasoning capabilities of large language models (LLMs) (Achiam et al., 2023; Yang et al., 2025), several works (Xu et al., 2024a; Khan et al., 2024a; You et al., 2024; Alrashedy et al., 2024; Wang et al., 2025; Li et al., 2025a) leverage LLMs or multimodal LLMs (MLLMs) to generate CAD models from natural language or other input modalities. Despite these advances, as shown in Figure 1, existing sequential representations remain limited to basic operations such as *sketch* and *extrude*. More sophisticated editing operations, including *chamfer* and *fillet*, are insufficiently supported. These operations refine designs by modifying existing geometry rather than creating new entities, and their correct execution demands explicit selections of geometric structures, an ability that current command sequence representations lack. Furthermore, the discretization of a continuous variable often suffer

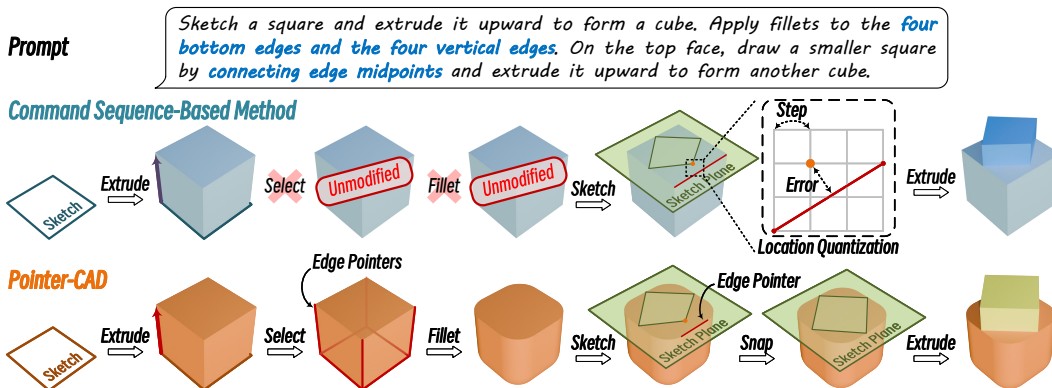

Figure 1: **Illustration of the strength of our proposed pointer-based command sequence compared to the command sequence-based CAD representation.** Command sequences suffer from the inability to refer to specific edges or faces, and discretization-induced quantization errors. In contrast, Pointer-CAD leverages edge pointers to directly refer to B-rep entities, enabling precise operations such as *sketch* snapping, thereby reducing quantization errors and faithfully following complex text instructions.

from quantization errors, which can disrupt otherwise continuous topological structures and hinder the effectiveness in practical applications.

To address these limitations, inspired by Pointer Networks (Vinyals et al., 2015), we propose a pointer-based representation that explicitly references B-rep elements (e.g., edges and faces). This design mimics an engineer's interaction with CAD software, enabling direct faces/edges selection and extending operations such as *chamfer* and *fillet*, which are crucial in industrial CAD modeling. Moreover, by snapping predictions to referenced B-rep elements indicated by these pointer, our representation can mitigate coordinate errors from regression or quantization. Building on the proposed pointer, we introduce a novel LLM-based text-to-CAD framework, **Pointer-CAD**. Unlike prior approaches that generate full CAD models in a single step, Pointer-CAD adopts a multi-step strategy by decomposing the model into distinct steps: at each step, the B-rep from previous steps and the textual description condition the LLM to generate the parametric subsequent components. Specifically, we extract geometric cues from B-rep faces and edges, construct a face-adjacency graph $\mathcal{G}$, and use graph neural networks (GNNs) (Scarselli et al., 2008) to aggregate local features from neighboring elements. Leveraging the reasoning capabilities of large language models, our framework outputs three complementary components, *Label Tokens, Value Tokens*, and *Pointer*, which can be directly translated into executable commands of CAD models. When an operation requires geometric dependency on a previously generated structure, such as applying a *chamfer* to an existing edge, the *Pointer* is activated to select the most feature matching candidate face or edge.

To facilitate the performance evaluation of text-to-CAD generation, we design a CAD annotation pipeline by leveraging Qwen2.5-VL (Bai et al., 2025) to generate high-level textual descriptions from multi-view CAD renderings. Building on the re-captioned OmniCAD dataset Xu et al. (2024a) and further extending it with *chamfer* and *fillet* operations, we obtain a total of 575,559 models. For fair comparison with existing baselines, we adopt the DeepCAD Wu et al. (2021) split from this re-captioned dataset. Our Pointer-CAD achieves strong performance on text-conditioned CAD generation, improving validity, command sequence accuracy, and geometric reconstruction fidelity. Notably, segment-level topological fidelity, measured by the Segment Error (SegE) metric, is reduced to the $10^{-3}$ scale—approximately $100\times$ lower than previous methods (Khan et al., 2024b; Govindarajan et al., 2025).

**To conclude, our contributions can be summarized as follows:** (1) A pointer-based CAD representation that enables selection of edges and faces, and making generation of advanced operations such as *chamfer* and *fillet* feasible to autoregressive methods and reducing the errors caused by quantization; (2) We introduce Pointer-CAD, an LLM-based text-to-CAD framework built on the proposed representation and employing a multi-step generation strategy, where each step is conditioned on both the textual description and the B-rep generated from previous steps; (3) Pointer-CAD outperforms baseline methods on text-conditioned generation in terms of validity, reconstruction quality, and topological consistency.

## 2 RELATED WORK

Boundary Representation (B-rep) (Ansaldi et al., 1985) uses a tree structure to organize vertices, edges, and faces in a hierarchical way. Several methods generate B-reps by progressively constructing these hierarchical structures (Nash et al., 2020; Guo et al., 2022; Jayaraman et al., 2022). Additionally, some recent approaches (Xu et al., 2024b; Liu et al., 2025) leverage latent spaces to encode the complex topology of B-reps. Based on these B-rep latent representation, CMT (Wu et al., 2025a) takes an effort to utilize a continuous autoregressive manner for B-rep generation. Although B-reps provide a direct representation of 3D models, the intricate relationships among elements make them challenging to generate.

Constructive Solid Geometry (CSG) (Foley, 1996) represents objects by combining primitive shapes through Boolean operations. Due to the non-uniqueness of CSG representations, researchers often employ unsupervised training methods (Sharma et al., 2018; Du et al., 2018; Kania et al., 2020). Recent works propose CSG-like representations (Ren et al., 2021) and learnable primitives (Yu et al., 2023; 2022) to improve generation quality. However, CSG methods struggle to represent curved surfaces such as rounded corners, limiting their capacity for complex geometries.

With the emergence of large-scale CAD datasets (Koch et al., 2019; Willis et al., 2021), researchers have begun to leverage deep models for CAD generation. DeepCAD (Wu et al., 2021) introduces a sequential representation by encoding design parameters as command sequences. SkexGen (Xu et al., 2022) proposes to integrate primitive hierarchy with command sequence for the autoregressive method. Some works (Ma et al., 2024; Zhang et al., 2025; Yu et al., 2025) further employ token-based diffusion models to generate command sequences. Recent research has explored the use of LLMs for CAD models and sketches generation from point clouds (Khan et al., 2024a), images (Chen et al., 2025; Niu et al., 2025; Wu et al., 2025b), and text (Khan et al., 2024b; Li et al., 2025c). CAD-MLLM (Xu et al., 2024a) proposes a multi-modal LLM framework that integrates these three modalities, and CAD-GPT (Wang et al., 2025) integrates images and text. Other methods represent the modeling process directly in plain text to simplify finetuning for LLMs (Zhang et al., 2024; Govindarajan et al., 2025; Guan et al., 2025; Li et al., 2025a). Additionally, some studies aim to enhance the geometric reasoning capabilities of LLMs through Chain-of-Thought (CoT) prompting (Li et al., 2025b) or hierarchical model structures (Dupont et al., 2024; Khan et al., 2024a). Existing command sequence representations lack explicit topological information, which remains a major challenge for autoregressive generation methods. A recent work (Fan et al., 2025) attempts to enable entity selection by labeling faces based on each operation and edges as intersections of faces. However, edges derived from face intersections may not be unique, leading to ambiguity in the selection function. A more robust solution for entity selection remains an open problem.

## 3 POINTER-BASED COMMAND SEQUENCES

Contemporary CAD software allows users to select operation targets directly on rendered geometry, e.g., clicking a plane for sketching or an edge for chamfering. In contrast, prior works (Wu et al., 2021; Khan et al., 2024b; Xu et al., 2024a; Govindarajan et al., 2025) represent operations only as numerical sequences, ignoring the geometric context from previous steps. As shown in Figure 1, this leads to two key issues: (i) operations like *chamfer* and *fillet* require explicit references to geometric entities and cannot be executed without corresponding edges or faces selection, making the operations unsupported; (ii) newly drawn curves often fail to snap to existing edges, or sketch planes can be misaligned with target faces. This issue arises because LLM-based sequential generation requires quantization of all continuous rotation and position parameters, introducing small errors that hinder precise geometric connectivity or alignment during sequential generation. Thus, motivated by Pointer Networks (Vinyals et al., 2015), we propose a novel pointer-based command sequence representation that explicitly integrates B-rep geometry into sequential modeling.

In our representation, each token belongs to one of three types: *Label Token*, *Value Token*, or *Pointer*. The *Label Token* carries explicit semantic information, indicating the type of an operation or a structural boundary in the sequence, as detailed in Table 1. The *Value Token* provides numerical data, such as coordinates or degrees. Notably, the continuous parameters are quantized into $2^q$ levels and expressed as $q$-bit integers. The *Pointer* is used to reference a face or an edge from the B-rep. Different operations are then defined by specific combinations and sequential order of these tokens. We decompose the entire CAD model construction process into a sequence of steps, each

Table 1: **Label Token Definitions.** This table provides a comprehensive list of all *Label Tokens* used in our command sequence representation, along with their semantic descriptions.

| Command | Description | Command | Description |
|---------|-------------|---------|-------------|
| *<ss>* | Start of sketch | *<sx>* | Start of curve |
| *<se>* | Start of extrusion | *<or>* | Orientation token group (Clockwise, Counter-Clockwise) |
| *<sc>* | Start of chamfer | *<dr>* | Direction token group (X+, X-, Y+, Y-, Z+, Z-) |
| *<sf>* | Start of fillet | *<bo>* | Boolean token group (New, Join, Cut, Intersect) |
| *<sp>* | Start of profile | ** | End of model (this step is the final step of the model) |
| *<sl>* | Start of loop | *<es>* | End of step (additional steps are required after this one) |

consisting of one of three fundamental operations: a sketch-extrude combination, a chamfer, or a fillet. And a CAD model is therefore represented by an ordered sequence of these operations.

**Sketch-extrude combination step.** Following prior works (Wu et al., 2021; Govindarajan et al., 2025; Khan et al., 2024b), we define a 2D sketch hierarchically: a sketch consists of faces, each bounded by one or more loops. A loop is formed by a sequence of primitive curves (lines or arcs) or a single circle, with consecutive curves sharing endpoints. The primitives are parameterized as: (i) $Line : (x, y)$, where $(x, y)$ defines the start point of a line; (ii) $Arc : (x, y, \alpha, o)$ which defines an arc with the start point $(x, y)$ and sweep angle $\alpha$. $o$ refers to the orientation flag (denoted as $<or>$); (iii) $Circle : (x, y, r)$, where $(x, y)$ is the center of an circle with a radius $r$.

For sketch plane selection, we replace the conventional parameterization using six values (three Euler angles and three translation parameters) with a pointer mechanism. This approach directly selects a target face from the B-rep representation to serve as the sketch plane. Once selected, a local 2D coordinate system is established on this plane, providing a consistent reference frame for all subsequent sketch operations (see Appendix A.2 for construction details). This pointer-based approach reformulates plane selection from a 3D rotation regression problem into a discrete selection over a finite set of candidate faces, reducing the search space and mitigating misalignment caused by inaccurate regression or quantization errors.

For the *extrude* operation, since the sketch plane has been determined, the operation can be simplified to be $E : (e_p, e_n, b)$, where $e_p, e_n$ denotes the extrusion distance towards the positive direction and negative direction of the sketch plane normal respectively, and $b$ (denoted as $<bo>$) is the type argument specifying the volume boolean type (e.g. *New*, *Join*, *Cut*, *Intersect*)

***Chamfer* or *fillet* operations step.** Mirroring the workflow in modern CAD software, both *chamfer* and *fillet* operations first require the selection of one or more target edges, followed by specifying a single numerical parameter. In our representation, a *chamfer* operation is expressed as $C : (\mathbf{p}, c)$ and a *fillet* as $F : (\mathbf{p}, f)$. Here, $\mathbf{p} = \{p_1, p_2, \ldots, p_n\}$ represents a set of pointers, with each pointer $p_i$ identifying a target edge from the B-rep. The parameters $c$ and $f$ denote the chamfer distance and fillet radius, which are applied uniformly across all selected edges.

## 4 METHOD

Building on the proposed pointer-based command sequence, we introduce Pointer-CAD, a framework that transforms natural language descriptions into 3D CAD models. In addition, we introduce an annotation pipeline and construct a new dataset to fully unleash the potential of Pointer-CAD. This section details the overall architecture, training objectives, and the annotation pipeline.

### 4.1 OVERALL ARCHITECTURE

As illustrated in Figure 2, unlike previous approaches that treat generation as a whole sequence, Pointer-CAD separates the process into multiple steps that are predicted sequentially in an autoregressive manner. Each prediction conditions on the text description and the B-rep geometry accumulated so far, ensuring global consistency and faithful design semantics. Pointer-CAD comprises three key components: a Multimodal Fusion Module that integrates text and B-rep geometry, an LLM for sequence generation, and a Vector Translation Module that converts command sequences into B-rep representations following the construction process described in Section 3.

#### 4.1.1 MULTIMODAL FUSION MODULE

The Multimodal Fusion Module takes tokenized text and B-rep geometry as inputs and integrates them to provide structured information for subsequent processing. The textual description is tok-

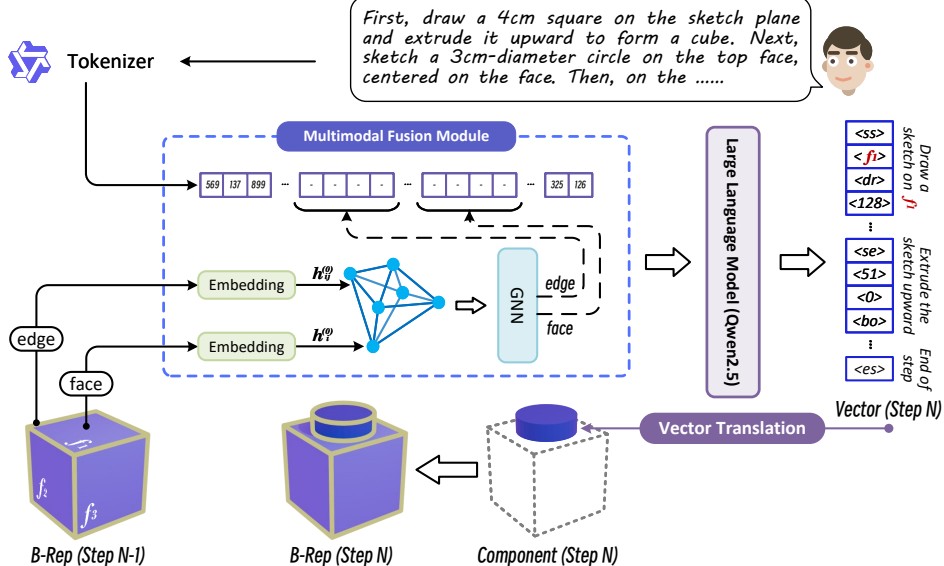

Figure 2: **Pointer-CAD Pipeline.** At each generation step, the full user prompt is tokenized, while the B-rep is updated with all geometry generated so far. A multimodal fusion module combines the textual prompt with the evolving B-rep, which is further encoded via a graph neural network over its faces and edges. The fused representation is then processed by a large language model to predict the vector for the current step, which is subsequently translated into geometry to update the B-rep.

enized once and reused across all steps, while the B-rep is updated after each operation; at the first step, it is empty, so the model conditions only on text.

**B-rep Encoder.** We represent the B-rep as an undirected face-adjacency graph $\mathcal{G}(V, E)$, where nodes correspond to faces and edges to shared boundaries. Following (Jayaraman et al., 2021; Yin et al., 2025), we construct the initial graph $\mathcal{G}$ by sampling geometric cues from the parametric domains of B-rep faces and edges. Each face is represented as $\mathcal{S}(u, v)$ and uniformly sampled on a 32×32 grid in the $(u, v)$ domain, with 3D coordinates, surface normals, Gaussian curvature, and visibility indicator concatenated as features. Similarly, each edge is parameterized as $\mathcal{C}(t)$ and uniformly sampled with 32 points, extracting the 3D coordinates, tangent and its reverse vector, and first-order derivative. Point-wise features are aggregated via average pooling and projected to a 128-d embedding, yielding node features $h_i^{(0)}$, and the edge feature $h_{ij}^{(0)}$, where $i, j$ are the indices of the faces for the initial graph $\mathcal{G}$. Further details are in the Appendix B.1.

**Graph Processing.** After obtaining the initial features, we take into account the structural properties of the B-rep and apply a $K$-layer Graph Neural Network (GNN)(Scarselli et al., 2008) to propagate information across the graph. Specifically, node features are updated by aggregating messages from their neighboring faces, while edge features require more nuanced handling. In B-reps, edges are not only incident to two adjacent faces but may also be indirectly related to other elements through shared vertices. To capture these dependencies, we update them using a Multi-Head Attention (MHA)(Vaswani et al., 2017) over all node features.

At the $k$-th layer, the updates are formulated as:

$$h_i^{(k)} = \phi^{(k)}\Big((1 + \epsilon^{(k)})\, h_i^{(k-1)} + \sum_{j \in \mathcal{N}(i)} f_\Theta(h_{ij}^{(k-1)}) \odot h_j^{(k-1)}\Big), \tag{1}$$

$$h_{ij}^{(k)} = \mathrm{MHA}\Big(Q = h_{ij}^{(k-1)},\ K, V = \{h_l^{(k-1)} \mid l \in \mathcal{V}\}\Big) + h_{ij}^{(k-1)}, \tag{2}$$

where $\phi^{(k)}$ denotes an MLP, $\epsilon^{(k)}$ is a learnable scalar, and $f_\Theta$ projects edge features into the node feature space. The resulting node and edge embeddings, $h_i^{(k)}$ and $h_{ij}^{(k)}$, are serialized into the LLM input via structured prompting: edge embeddings are wrapped as $<brep\_edge\_start>$ *edge embedding* $<brep\_edge\_end>$ and face embeddings as $<brep\_face\_start>$ *face embedding* $<brep\_face\_end>$, providing the LLM with explicit structural cues to distinguish B-rep components.

### 4.1.2 SUPERVISED FINETUNING OF LARGE LANGUAGE MODELS

LLMs have demonstrated strong understanding and reasoning capabilities over structured inputs. In Pointer-CAD, we adopt Qwen2.5 (Team, 2024) as the backbone LLM and leverage Low-Rank Adaptation (LoRA) (Hu et al., 2022) to reduce trainable parameters. To align with our representation system, we append two separate fully connected layers to the final hidden state of the LLM: one predicts the *Label Token* and *Value Token*, while the other predicts the *Pointer*. Outputs are then translated into executable command sequences according to the rules detailed in the Appendix A.1.

**Pointer-based Referencing.** In the pointer-enabled setting, the LLM predicts a *Pointer* to select the target face or edge from a set of candidates. We denote the complete set of faces (including the three base planes: Right, Front, and Top) and edges as $\mathcal{S}_f$ and $\mathcal{S}_e$, respectively. The ground-truth target is defined as a subset of these candidates, potentially containing more than one element, because geometric relationships (e.g., coplanar faces, collinear edges) naturally admit multiple valid references. Precise definitions of these geometric special cases are provided in the Appendix A.3. Formally, for the $m$-th predicted face pointer, we define the ground-truth set as $\mathcal{P}_m \subseteq \mathcal{S}_f$, and $\mathcal{N}_m = \mathcal{S}_f \setminus \mathcal{P}_m$ is negative. Similarly, $\mathcal{P}_n \subseteq \mathcal{S}_e$ and $\mathcal{N}_n = \mathcal{S}_e \setminus \mathcal{P}_n$ for the $n$-th predicted edge pointer. Each candidate uses its initial feature: $h_i^{(0)}$ for the $i$-th face in $\mathcal{S}_f$, and $h_{ij}^{(0)}$ for the edge shared by the $i$-th and $j$-th faces in $\mathcal{S}_e$, with three base planes encoded as distinct learnable 128-d embeddings, aligning with features in both $\mathcal{S}_f$ and $\mathcal{S}_e$. To predict a face or edge pointer, the LLM outputs a 128-d vector, and then matched to the candidate geometric element with highest cosine similarity.

## 4.2 TRAINING OBJECTIVE

Based on the structure of the command sequence, our training objective is to jointly predict the correct token value and referenced pointer representation.

**Label and Value Token Prediction.** The prediction of both *Label Tokens* and *Value Tokens* is formulated as a classification task. Given the constrained output space, we employ a cross-entropy loss with label smoothing, defined as:

$$\mathcal{L}_v = -\sum_{i=1}^{N} \left[ (1-\alpha) \cdot \delta_{i,y} + \frac{\alpha}{N-1} \cdot (1-\delta_{i,y}) \right] \log p_i, \tag{3}$$

where $\delta_{i,y}$ is the Kronecker delta (1 if $i = y$, 0 otherwise), $y$ is the correct class, $N$ is the number of classes, $\alpha$ is the label smoothing factor, and $p_i$ is the predicted probability of class $i$, obtained via softmax over the model logits.

**Pointer Prediction.** Pointer prediction is cast as a regression task. Since multiple valid pointers may exist simultaneously, we adopt a contrastive-style loss:

$$\mathcal{L}_p = -\frac{1}{|\mathcal{P}| + |\mathcal{N}|} \left[ \sum_{j \in \mathcal{P}} \log \left( \sigma \left( \frac{\cos(p, c_j)}{\tau} \right) \right) + \sum_{j \in \mathcal{N}} \log \left( 1 - \sigma \left( \frac{\cos(p, c_j)}{\tau} \right) \right) \right], \tag{4}$$

where $\mathcal{P}$ and $\mathcal{N}$ denote the sets of valid and invalid candidates, $p$ is the predicted pointer embedding, $c_j$ is the embedding of candidate $j$, $\sigma$ is the sigmoid function, and $\tau$ is a learnable temperature.

**Overall Objective.** The overall loss is a weighted sum of the two objectives:

$$\mathcal{L} = \lambda_v \cdot \mathcal{L}_v + \lambda_p \cdot \mathcal{L}_p, \tag{5}$$

where $\lambda_v$ and $\lambda_p$ are hyperparameters controlling the relative contributions of these two components.

## 4.3 ANNOTATION PIPELINE

As shown in Figure 3, we render four multi-view images per model using Blender and use Qwen2.5-VL (Bai et al., 2025) to generate a one-word label and single-sentence caption for global shape understanding. For each sketch plane, six images are rendered from different angles with the plane highlighted in red, and Qwen2.5-VL provides a macro-level spatial location description. These visual and textual annotations offer a more comprehensive understanding of both the model geometry and the sketch plane location. Following Text2CAD (Khan et al., 2024b), we further convert the raw JSON files into a minimal and human-readable JSON format, enhanced with textual descriptions to improve interpretability. We also employ Qwen2.5 (Yang et al., 2024) to generate natural language modeling instructions, with all dimension parameters wrapped in <v> tags.

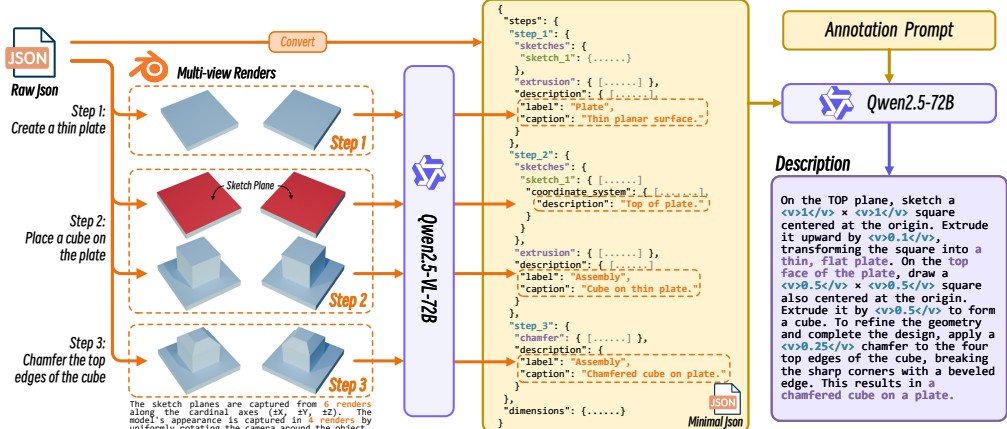

Figure 3: **Dataset construction pipeline.** Raw JSON files are converted into a minimal JSON format containing only annotation-relevant elements. Sketch planes and models are rendered, and Qwen2.5-VL generates textual descriptions that are integrated into the minimal JSON. Finally, Qwen2.5 produces step-by-step natural language instructions, with dimension parameters wrapped in special tags for future data augmentation.

Building on OmniCAD (Xu et al., 2024a), we apply its augmentation strategy by splitting full models into sub-models by intermediate stages, annotate all sub-models via our pipeline, and denote the re-captioned dataset as Recap-OmniCAD. Since OmniCAD originally lacks *chamfer* and *fillet* operations, which were skipped during augmentation, we reintegrate them and extend the dataset, yielding Recap-OmniCAD⁺. Dataset statistics and annotation prompts are reported in the Appendix C.

## 5 EXPERIMENTS

### 5.1 EXPERIMENTAL SETUP

**Datasets.** To demonstrate the effectiveness of Pointer-CAD and ensure fair comparison, we additionally annotate a subset of the DeepCAD dataset, denoted as Recap-DeepCAD, which contains 176,439 CAD models. For the ablation studies, we train on the Recap-OmniCAD⁺ dataset by default. Following OmniCAD's evaluation protocol, no data augmentation is applied to the test set, which consists of 13,971 models.

**Implementation Details.** We adopt Qwen2.5-0.5B (Yang et al., 2024) as the base LLM for implementing Pointer-CAD, unless otherwise specified. The model is trained for 10 epochs on 16 NVIDIA H800 GPUs. Further details are provided in the Appendix B.3.

**Metrics.** We evaluate the quality of generated CAD models using model validity, reconstruction quality and topological accuracy metrics following Text2CAD (Khan et al., 2024b) and CADmium (Govindarajan et al., 2025). Specifically, we report Invalidity Ratio (IR), F1 score, Chamfer Distance (CD), Segment Error (SegE), Dangling Edge Length (DangEL), Self-Intersection Ratio (SIR), and Flux Enclosure Error (FluxEE). The F1 score reflects command sequence accuracy, while CD captures geometric reconstruction fidelity. SegE, DangEL, and SIR measure different aspects of topological soundness, and FluxEE quantifies the deviation from a watertight, enclosed solid. All reported CD and FluxEE values are scaled by $10^3$ for clarity.

### 5.2 COMPARISON ON TEXT CONDITIONED CAD GENERATION

We involve two open-source text-to-CAD baselines for comparison: Text2CAD (Khan et al., 2024b) and CADmium (Govindarajan et al., 2025). In addition, we adapt DeepCAD (Wu et al., 2021) for text-conditioned generation by reusing its pretrained latent-space decoder, which was trained on the DeepCAD dataset, and training a new encoder to map text inputs into the corresponding latent vectors. To ensure fair comparison, all the baseline methods and Pointer-CAD are trained on Recap-DeepCAD. As shown in Table 2(a), Pointer-CAD-1.5B achieves the best IR, sketch operation F1, CD, while Pointer-CAD-0.5B attains the best performance on the remaining metrics. Notably, our method achieves a SegE over 100× smaller than all baselines, demonstrating that the proposed pointer mechanism effectively mitigates discontinuities caused by even tiny quantization errors, which can separate newly drawn parts from existing geometry. Moreover, our method attains superior overall performance with a 0.5B model size compared to the 7B-LLM-based CADmium.

Table 2: **Quantitative comparison on different datasets.** (a) Recap-DeepCAD dataset: Pointer-CAD (0.5B/1.5B) achieves the highest operation F1 scores and lowest CD errors, outperforming other baselines and larger LLM-based method CADmium-7B. (b) Recap-OmniCAD$^+$ dataset: Pointer-CAD uniquely supports *chamfer* and *fillet* operations with high accuracy, while other methods fail, and further demonstrates superior geometric fidelity and topology quality.

| | (a) Recap-DeepCAD | | | | | | | (b) Recap-OmniCAD$^+$ | | | | | | |
| | | | CADmium | | | Pointer-CAD | | | | | CADmium | | | Pointer-CAD | |
| Metric | DeepCAD | Text2CAD | 1.5B | 3B | 7B | 0.5B | 1.5B | DeepCAD | Text2CAD | 1.5B | 3B | 7B | 0.5B | 1.5B |
|---|---|---|---|---|---|---|---|---|---|---|---|---|---|---|
| IR ↓ | 39.23 | 30.16 | 42.84 | 36.34 | 31.79 | 15.02 | **8.80** | 43.87 | 34.60 | 48.11 | 43.74 | 38.76 | 25.37 | **19.15** |
| Line F1 ↑ | 80.14 | 88.12 | 85.47 | 82.25 | 85.13 | 97.70 | **98.73** | 82.40 | 86.37 | 81.26 | 81.81 | 82.78 | 94.37 | **95.79** |
| Arc F1 ↑ | 31.41 | 45.19 | 19.35 | 20.44 | 25.68 | 85.70 | **95.14** | 27.97 | 37.04 | 19.61 | 25.47 | 23.87 | 67.62 | **74.98** |
| Circle F1 ↑ | 79.04 | 87.03 | 75.64 | 72.66 | 74.94 | 98.27 | **98.66** | 64.11 | 73.41 | 64.88 | 64.52 | 67.35 | 95.61 | **96.03** |
| Extrusion F1 ↑ | 92.34 | 98.53 | 92.50 | 88.50 | 90.75 | **99.67** | 99.61 | 90.21 | 98.11 | 91.68 | 88.42 | 90.62 | **99.22** | 99.20 |
| Chamfer F1 ↑ | - | - | - | - | - | - | - | - | - | - | - | - | 89.74 | **94.32** |
| Fillet F1 ↑ | - | - | - | - | - | - | - | - | - | - | - | - | 82.54 | **89.85** |
| CD mean ↓ | 37.47 | 17.48 | 11.51 | 12.22 | 10.53 | 3.81 | **2.58** | 39.89 | 19.80 | 19.35 | 17.93 | 13.84 | 5.49 | **2.86** |
| CD median ↓ | 12.56 | 3.38 | 0.57 | 0.47 | 0.44 | 0.54 | **0.30** | 12.71 | 3.52 | 0.98 | 0.99 | 0.88 | 0.53 | **0.34** |
| SegE ↓ | 0.53 | 0.44 | 0.47 | 0.64 | 1.21 | **1.33e-3** | 1.38e-3 | 0.84 | 0.53 | 0.73 | 0.89 | 1.78 | 1.41e-3 | **1.01e-3** |
| DangEL ↓ | 1.80 | 0.71 | 3.27 | 3.88 | 5.33 | **0.16** | 0.20 | 4.80 | 1.67 | 3.32 | 4.79 | 4.95 | 0.28 | **0.26** |
| SIR ↓ | 0.15 | 0.07 | 0.10 | 0.13 | 0.20 | **0.01** | 0.02 | 0.19 | 0.11 | 0.13 | 0.15 | 0.21 | 0.02 | **0.02** |
| FluxEE ↓ | 25.85 | 17.75 | 38.63 | 29.73 | 32.22 | **1.99** | 3.02 | 40.06 | 27.54 | 36.47 | 31.03 | 36.30 | 3.57 | **3.37** |

Qualitative comparison can be observed from Figure 4. Baseline methods frequently produce defective CAD models, exhibiting issues such as overly thin surfaces or incorrect spatial arrangement of internal structures. In contrast, our method explicitly incorporates geometric information from B-rep into the modeling process, leading to significantly improved structural accuracy. Overall, our pointer-based command sequence representation, together with the training strategy, shows strong compatibility with autoregressive models.

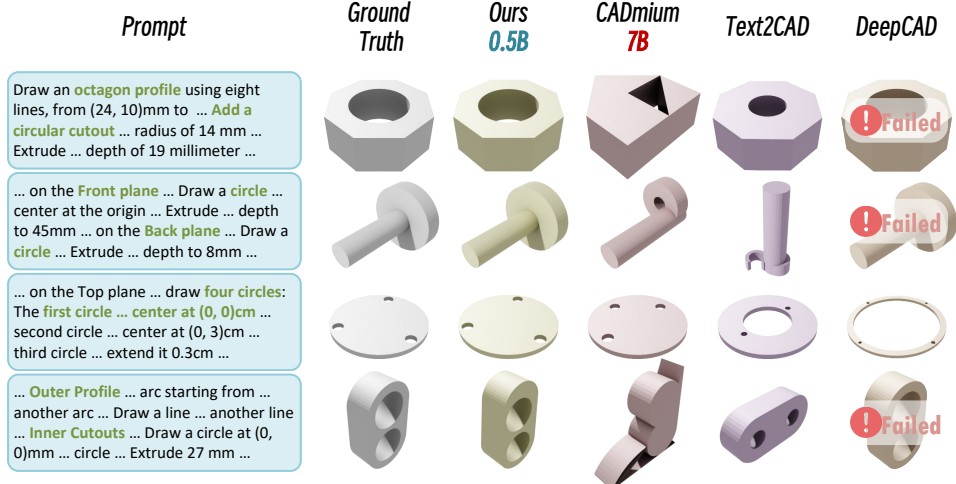

Figure 4: **Qualitative performance comparison on Recap-DeepCAD dataset.** Our method (Pointer-CAD-0.5B) consistently produces accurate and faithful geometry aligned with the ground truth, while competing methods often fail to capture geometric details or even collapse entirely. Among LLM-based methods, Pointer-CAD achieves superior results despite its significantly smaller model size compared to CADmium.

### 5.3 VALIDATION OF SUPPORT FOR *Chamfer* AND *Fillet* OPERATIONS

To assess our model's capability on *chamfer* and *fillet* operations, we train it on the Recap-OmniCAD$^+$ dataset, which includes these two additional operation types. Baseline methods, which do not support *chamfer* and *fillet*, are trained on the Recap-DeepCAD dataset instead. As illustrated in Figure 5, existing methods often fail to execute these operations correctly, producing invalid results. In contrast, our approach faithfully reconstructs these operations, achieving higher F1 scores, better geometric accuracy, and superior topology quality compared to all baselines, as in Table 2(b).

### 5.4 ABLATION ON THE PARAMETER QUANTIZATION LEVEL

As discussed in Section 3, command sequences discretize continuous parameters (e.g., coordinates, extrusion distances, angles) into $2^q$ levels. To evaluate the effect of different quantization levels on our model's performance, we train Pointer-CAD-0.5B on the Recap-OmniCAD$^+$ dataset with levels of 64, 256, and 1024. As shown in Table 3, models across different quantization levels achieve comparable IR, while the 1024-level setting notably attains the highest F1 score across all operation

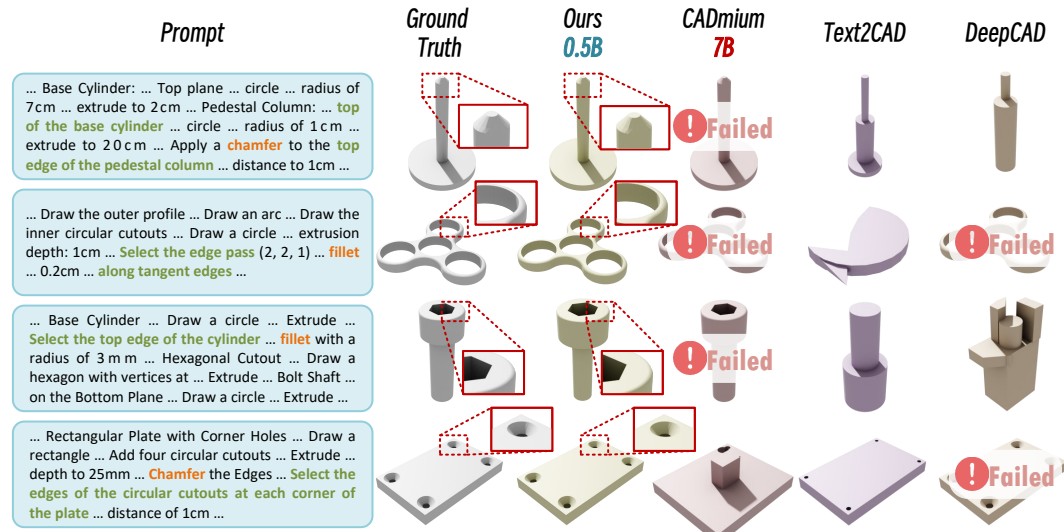

Figure 5: **Qualitative performance comparison on Recap-OmniCAD⁺ dataset.** Our method (Pointer-CAD-0.5B) accurately recovers detailed structures that closely match the ground truth for complex CAD models involving *chamfer* or *fillet* operations. In contrast, competing methods often miss fine-grained features even when they do not fail entirely, resulting in outputs that deviate from the user's intended design.

Table 3: **Performance of Pointer-CAD-0.5B under different integer quantization levels on Recap-OmniCAD⁺.** While IR remain comparable across resolutions, the 1024 setting achieves the highest command sequence F1 score and best results in CD, demonstrating improved fidelity in geometric reconstruction.

| Quantization Level | 64 | 256 | 1024 |
|---|---|---|---|
| IR ↓ | 25.45 | **25.37** | 25.79 |
| Line F1 ↑ | 93.00 | 94.37 | **94.94** |
| Arc F1 ↑ | 63.35 | 67.62 | **72.06** |
| Circle F1 ↑ | 94.81 | 95.61 | **96.16** |
| Extrusion F1 ↑ | 99.26 | 99.22 | **99.44** |
| Chamfer F1 ↑ | 90.75 | 89.74 | **91.16** |
| Fillet F1 ↑ | 77.96 | 82.54 | **87.93** |
| CD mean ↓ | 5.92 | 5.49 | **4.93** |
| CD median ↓ | 0.77 | 0.53 | **0.52** |

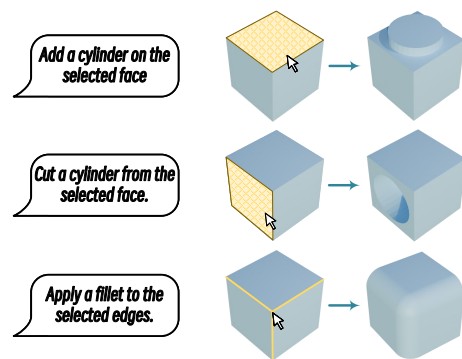

Figure 6: **Illustration of our interactive editing functionality.** Users can directly click on a face or edge of the CAD model and provide a text prompt to specify the desired operation.

types in the command sequence evaluation. It also achieves competitive results in CD, highlighting that finer quantization levels yield better geometric fidelity.

## 5.5 EXTENSION TO CLICK INTERACTION EDITING APPLICATION

Since our proposed pointer-based command sequence allows entity selection at each step, we extend the model with token concatenation to incorporate user-interactive selections alongside text instructions, enabling an immersive editing experience. As illustrated in Figure 6, users can interactively select faces or edges on the current B-rep to explicitly specify the operation target, enabling more precise and intuitive editing through direct manipulation in conjunction with text instructions.

## 6 CONCLUSION

In this work, we present Pointer-CAD, an LLM-based framework supporting *chamfer* and *fillet* operations for generating complex CAD models. To ensure geometric accuracy and enable entity selection, we introduce a pointer-based command sequence that explicitly incorporates B-rep geometry, allowing the model to reference existing faces and edges. We further enhance training with a dedicated dataset annotation pipeline. Extensive experiments show that Pointer-CAD can produce models with higher topological accuracy and geometric fidelity conditioned on the input text.

ETHICS STATEMENT

The models and annotations are used solely for research purposes, and our pipeline does not produce outputs that could compromise privacy or safety. Meanwhile, we also acknowledge potential misuse in automated CAD generation, such as intellectual property concerns. We encourage responsible use of our work in research and industry, adhering to ethical standards and respecting ownership of CAD designs.

REPRODUCIBILITY STATEMENT

We provide detailed training information in Appendix D. The pretrained LLM weights used, including Qwen2.5 and Qwen2.5-VL, are publicly available. Annotated prompts are provided in Appendix C.2. The datasets used, DeepCAD (Wu et al., 2021) and OmniCAD (Xu et al., 2024a), are publicly accessible. The extended content in OmniCAD is derived from ABC (Koch et al., 2019) using the index split provided by OmniCAD, which can be publicly accessed. Finally, we will release all training and inference code along with the re-captioned text descriptions of OmniCAD to facilitate reproducibility.

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

# Supplementary Material

In these supplementary materials, we provide the following:

- Details of the proposed pointer-based representation, including sketch plane selection method, specific vector translation rules, and definitions of geometric special cases in pointer-based referencing;
- Details of the training framework, covering the B-rep encoder, implementation details of the autoregressive decoder, and the training objective;
- Visualization of some dataset cases, the prompts used for annotation, and dataset statistics;
- Detailed Implementation Details;
- Discussions on future directions;
- Role of LLM usage in our work.

## A   DETAILS OF THE POINTER-BASED REPRESENTATION

This section elaborates on the implementation logic of the pointer-based representation and the methodology for sketch plane selection.

### A.1   SPECIFIC VECTOR TRANSLATION RULES

Each token is classified as one of three types: *Label Token*, *Value Token*, or *Pointer*. To simplify the model architecture, we assign non-overlapping integer ranges to label and value tokens, allowing them to be decoded by a single prediction head. However, since a pointer is a reference to a geometric entity rather than a simple value, it requires a separate prediction head for decoding. To distinguish pointers from label and value tokens, we reserve two specific integer values within the label/value token space. When the model predicts one of these integers, it signals that the current token is a pointer. These two integers also represent the pointer's state: as shown in Table 4, $<pe>$ indicates an enabled pointer that references an edge or face, whereas $<pd>$ signifies a disabled (inactive) pointer. Specifically, for $<nv>$ and $<ag>$, we quantize all continuous parameters into $2^q$ levels and express them using $q$-bit integers. And then the value is normalized to the expected range.

Table 4: **Special Token Definitions.** This table provides a comprehensive list of all Special Tokens used in our command sequence representation, along with their semantic descriptions.

| Notation | ID | Type | Description |
|---|---|---|---|
| $$ | 1 | Label Token | End of model (this step is the final step of the model) |
| $<es>$ | 2 | Label Token | End of step (additional steps are required after this one) |
| $<ss>$ | 3 | Label Token | Start of sketch |
| $<se>$ | 4 | Label Token | Start of extrusion |
| $<sc>$ | 5 | Label Token | Start of chamfer |
| $<sf>$ | 6 | Label Token | Start of fillet |
| $<sp>$ | 7 | Label Token | Start of profile |
| $<sl>$ | 8 | Label Token | Start of loop |
| $<sx>$ | 9 | Label Token | Start of curve |
| $<pe>$ | 10 | Pointer | Pointer to an Edge or Face |
| $<pd>$ | 11 | Pointer | Empty pointer |
| $<or>$ | $\{12, 13\}$ | Label Token | Orientation (Clockwise, Counter-Clockwise) |
| $<dr>$ | $[14, 19]$ | Label Token | Direction (X+, X-, Y+, Y-, Z+, Z-) |
| $<bo>$ | $[20, 23]$ | Label Token | Boolean (New, Join, Cut, Intersect) |
| $<nv>$ | $[24, 24 + 2^q]$ | Value Token | Normalized to $[0, 1]$ and then quantize to $2^q$ level |
| $<ag>$ | $[24, 24 + 2^q]$ | Value Token | Normalized to angle $[0°, 360°)$ and then quantize to $2^q$ level |

Based on the notation in Table 4, we define the translation rules for each command as shown in Table 5. A CAD model is represented as a sequence of valid sequences, with only the last valid sequence is end with $$.

Table 5: **Sequence definitions.** Each sequence is defined by a specific combination of commands. The superscript [ $^+$ ] denotes that the element appears one or more times, while the symbol [ / ] indicates that one of the alternatives can be chosen.

| Notation | Sequence | Description |
|---|---|---|
| [P] | $<nv>$ $<nv>$ $<pe\,/\,pd>$ | 2D point $(x, y)$, snapped to reference or placed freely |
| [L] | $<sx>$ [P] | Line starting from a 2D point $(x, y)$ |
| [C] | $<sx>$ [P] $<nv>$ | Circle with center at $(x, y)$ and radius $r$ |
| [A] | $<sx>$ [P] $<ag>$ $<or>$ | Arc starting from $(x, y)$ with angle $\alpha$ and orientation |
| [Loop] | $<sl>$ [L / C / A]$^+$ | Closed loop composed of multiple curves |
| [Profile] | $<sp>$ [Loop]$^+$ | 2D Region defined by one or more loops |
| [CS] | $<dr>$ [P] $<ag>$ $<nv>$ | 2D coordinate system in 3D space |
| [Sketch] | $<ss>$ $<pe>$ [CS] [Profile]$^+$ | Sketch on a plane specified by pointer |
| [Extrude] | $<se>$ $<nv>$ $<nv>$ $<bo>$ | Extrude operation with depth and Boolean type |
| [EPart] | [Sketch]$^+$ [Extrude] | Solid part constructed by extrusion |
| [Chamfer] | $<sc>$ $<nv>$ $<pe>^+$ | Chamfer operation on referenced edges |
| [Fillet] | $<sf>$ $<nv>$ $<pe>^+$ | Fillet operation on referenced edges |
| **[VSeq]** | **[EPart / Chamfer / Fillet]** $<es\,/\,em>$ | **Valid sequence** |

## A.2 SKETCH PLANE SELECTION

As defined in the [Sketch] notation, a sketch plane is specified by a *Pointer* to a face and a 2D coordinate system [CS]. The construction process, illustrated in Figure 7, unfolds in three main steps: First, a base plane is established by selecting a face with the *Pointer*, as shown in Figure 7a. The resulting sketch plane is coplanar with this face. Second, a local coordinate system $U'V'W'$ is constructed on this plane. The normal axis, $W'$, is aligned with the face normal that has a positive dot product with a world axis direction, $n$, specified by the *Label Token* $<dr>$. The primary in-plane axis, $U'$, is determined by projecting an auxiliary direction, $d$ (listed in Table 6), onto the sketch plane. The second in-plane axis, $V'$, is then derived using the right-hand rule, completing the orthogonal basis $U'V'W'$. As depicted in Figure 7b, the origin of this system is defined by projecting a point $P$ from a world coordinate plane onto the sketch plane along the direction $n$. Finally, as shown in Figure 7c, the final sketch coordinate system $UVW$ is obtained by applying a counterclockwise in-plane rotation to $U'V'W'$ about the $W$-axis. An optional scaling factor may also be applied to mitigate quantization errors.

Table 6: **Direction mapping.** In command $<dr>$, each symbol corresponds to a primary direction and its auxiliary direction.

| Symbol | Direction | Auxiliary Direction |
|---|---|---|
| 14 | X+ | Y+ |
| 15 | X- | Z+ |
| 16 | Y+ | Z+ |
| 17 | Y- | X+ |
| 18 | Z+ | X+ |
| 19 | Z- | Y+ |

## A.3 GEOMETRIC SPECIAL CASES IN POINTER REFERENCING

While a pointer is generally intended to reference a single, unique geometric entity (i.e., an edge or a face), this one-to-one correspondence breaks down in certain "geometric special cases." These cases occur when multiple entities are geometrically equivalent from a modeling standpoint, such as coplanar faces or collinear edges. In such scenarios, selecting any one of these equivalent entities would result in the same final geometry. Therefore, the ground truth for a pointer is not a single ob-

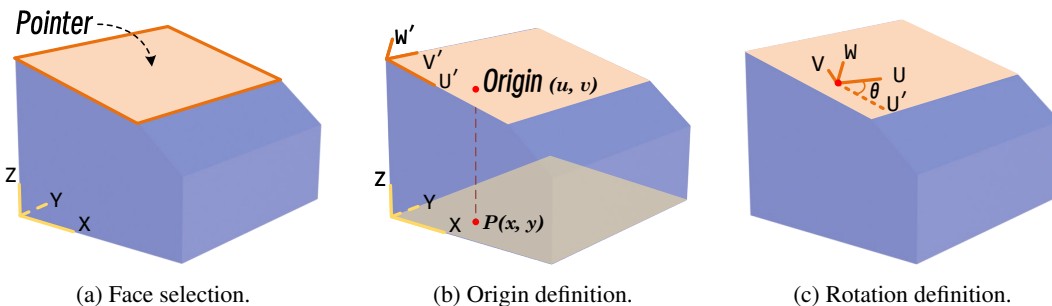

(a) Face selection.      (b) Origin definition.      (c) Rotation definition.

Figure 7: **Sketch coordinate system construction.** The sketch plane, axes, origin, and rotation are defined step by step to form the local coordinate system $UVW$.

ject but rather a set of valid candidates. This section provides precise definitions for these geometric special cases.

**Coplanar-Adjacent Faces.** A face pointer selects a base face to define a sketch plane. If two or more faces are coplanar (i.e., they lie on the same geometric plane), selecting any of them will result in the same sketch plane definition. Therefore, all faces within such a coplanar group are considered valid candidates for the face pointer.

**Collinear-Connected Edges.** Snapping a sketch point to an existing edge requires an edge pointer. If other edges are collinear with the target edge, pointing to any of them will produce the same snapping result. Therefore, all edges within such a collinear group are considered valid candidates for the edge pointer.

## B  DETAILS OF THE TRAINING FRAMEWORK.

### B.1  B-REP ENCODER

For each B-rep edge, we uniformly sample 32 points along its parametric curve in 3D space and extract four quantities at each location: point coordinates, tangent and its reverse vector, and first-order derivatives. Each is represented as a 3D vector, and their concatenation yields a 12-dimensional feature per sample. Collecting all samples forms an edge feature tensor of shape $32 \times 12$, which serves as input for edge embedding.

For each B-rep face, we uniformly sample its parametric $(u, v)$ domain to construct a regular UV grid of size $32 \times 32$. At each grid point, we compute the 3D coordinates, unit surface normal, Gaussian curvature, and a binary visibility mask (set to 1 for interior or boundary samples and 0 otherwise). Concatenating these quantities channel-wise gives an 8-dimensional feature per location, i.e., $3 + 3 + 1 + 1$, producing a face tensor of shape $32 \times 32 \times 8$.

Node embeddings are obtained by applying 2D convolutions to the face tensor, expanding it to 256 channels, followed by global adaptive average pooling and a linear projection to a 128-dimensional vector, denoted $h_i^{(0)}$. Similarly, edge embeddings are obtained by applying 1D convolutions to the edge tensor, expanding it to 256 channels, followed by global adaptive average pooling and a linear projection to a 128-dimensional vector, denoted $h_{ij}^{(0)}$. Thus, the graph $\mathcal{G}$ is initialized with node features $h_i^{(0)}$ and edge features $h_{ij}^{(0)}$ for downstream processing.

### B.2  IMPLEMENTATION DETAILS OF THE AUTOREGRESSIVE DECODER

To translate the output of the LLM into our defined command sequence, we process the last hidden state from the model's transformer decoder at each autoregressive decoding step. We employ a dual-head architecture to decode the hidden state into the appropriate token type.

The first head, which we refer to as the **Label/Value Head**, is a linear layer responsible for predicting both *Label Tokens* and *Value Tokens*. Its output dimension is size(Label Token) $+ 2 + 2^q$, which aligns with the tokenization scheme described previously:

- size(Label Token) corresponds to the vocabulary size of all possible *Label Tokens* defined in Table 4.

- 2 represents the two special tokens, $<pe>$ and $<pd>$, that signal a pointer's state. When the model predicts one of these, it indicates that the current token is a pointer, and the output from the second head should be used.

- $2^q$ represents the quantized bins for all continuous *Value Tokens*, such as those for $<nv>$ and $<ag>$.

The second head, the **Pointer Head**, is another linear layer specifically designed for decoding pointers. This head's output is a 128-dimensional vector. When the Label/Value Head predicts a pointer state, this 128-dimensional vector is used to perform a similarity search (via cosine similarity) against the 128-dimensional embeddings of all candidate geometric entities (faces and edges) generated by the B-rep encoder. The entity with the highest similarity score is selected as the pointer's reference. This mechanism allows the model to dynamically ground its generation in the existing B-rep geometry.

### B.3 Details of Training Objective

**Pointer Prediction.** Following CLIP Radford et al. (2021), we employ a learnable temperature parameter $\tau$ to control the scale of the logits in the loss computation. The parameter is initialized to 0.07, following Wu et al. (2018). To improve training stability, we reparameterize $\tau$ as its reciprocal $s = 1/\tau$ and optimize $\log s$ during training, with $s$ clipped to $s \leq 100$ to avoid excessive scaling of the logits. The learning rate for $s$ is set to $lr_s = 0.1 \times lr$, and weight decay is not applied during its optimization.

**Overall Objective.** The overall training objective $\mathcal{L}$ combines the cross-entropy loss for label/value tokens ($\mathcal{L}_v$) and the contrastive loss for pointer tokens ($\mathcal{L}_p$). The final loss is a weighted sum of these two components, controlled by hyperparameters $\lambda_v$ and $\lambda_p$. In all our experiments, we set $\lambda_v = 0.5$ and $\lambda_p = 0.5$ to give them equal weight.

## C Details of the Dataset

### C.1 Dataset Visualization

Figure 8 presents several representative samples from the Recap-OmniCAD$^+$ dataset, showcasing a wide spectrum of model complexity and diversity. As illustrated, our dataset contains a rich variety of models that not only feature complex geometric details such as fillets and chamfers but also exhibit diverse topological structures like holes, pockets, and multi-body components.

### C.2 Details of Annotation Prompts

In our dataset construction process, we employ a multi-step approach to generate rich and detailed annotations for each CAD model. First, we utilize the Qwen2.5-vl-72B model to generate a visual description of the model's appearance, using the prompt shown in Figure 11. Next, we use the same model to describe the relative position of the sketch plane within the model, guided by the prompt in Figure 12. To ensure a clear and accurate understanding for the model, we dynamically replace the placeholders in the prompt with the actual sketch plane surface normal vector and facing direction for each CAD model.

The resulting annotations are then combined with modeling parameters extracted from the raw JSON file to create a structured "minimal JSON," as illustrated in Figure 13. This minimal JSON, along with the prompt shown in Figure 14, is then passed to Qwen2.5-72B-Instruct to generate a final natural language description of the modeling process.

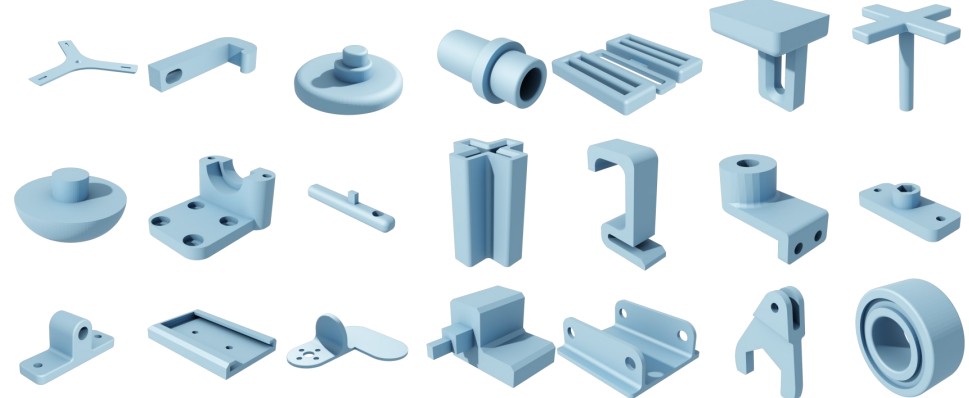

Figure 8: **Representative samples from the Recap-OmniCAD⁺ dataset.** The figure displays a range of models with varying complexity, from simpler parts with basic features to intricate components incorporating numerous fillets, chamfers, and complex sketches.

## C.3 DATASET STATISTICS

We provide a statistical analysis of our dataset in Figure 9 and Figure 10.

Figure 9 illustrates the distribution of modeling operations. Notably, Recap-OmniCAD⁺ includes *chamfer* and *fillet* operations, which are absent in the original OmniCAD. The reintegration of these features results in a higher count for all operation types in Recap-OmniCAD⁺ compared to Omni-CAD.

In Pointer-CAD, the command sequence of a complete CAD model is decomposed into three types of operations: sketch–extrude combinations, chamfers, and fillets. Figure 10 presents the statistics of our dataset according to this decomposition. The inclusion of *chamfer* and *fillet* operations increases the overall complexity and the average number of steps required to construct a model. This is reflected in the distribution, where Recap-OmniCAD⁺ has a slightly lower count of models with a single operation but a consistently higher count for models requiring more than one operation compared to OmniCAD.

Furthermore, both figures highlight that OmniCAD and Recap-OmniCAD⁺ are significantly more complex than DeepCAD. They feature a greater total number of operations and a higher proportion of models requiring a large number of construction steps. This demonstrates that our datasets are more challenging and better reflect the complexity of real-world CAD modeling tasks.

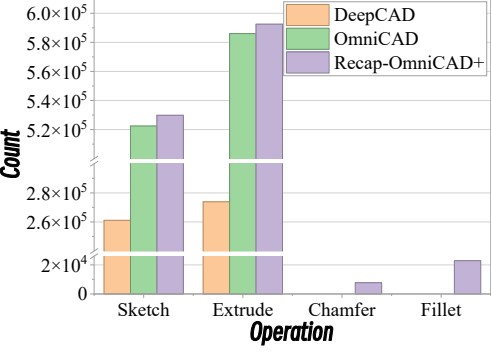

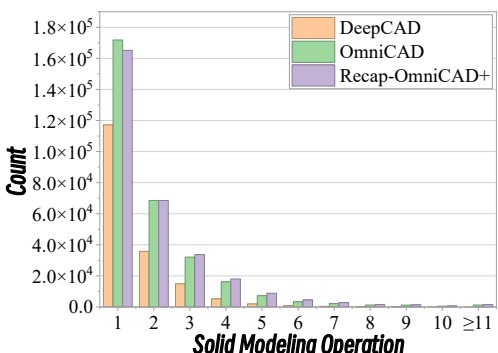

Figure 9: **Distribution of modeling operations across datasets.** The figure illustrates the total count of each modeling operation type for the DeepCAD, OmniCAD, and Recap-OmniCAD⁺ datasets.

Figure 10: **Distribution of modeling steps per model.** The figure compares the number of solid modeling operations required per model across the datasets.

## D DETAILS OF IMPLEMENTATION DETAILS

For the default 0.5B model setting, the entire training process requires approximately 23 hours on 16 NVIDIA H800 GPUs. We use the AdamW optimizer (Loshchilov & Hutter, 2017) with a learning rate of $1 \times 10^{-4}$ and a linear decay schedule. For LoRA, the dropout rate is set to 0.1. We use a micro-batch size of 9 with 2 gradient accumulation steps per GPU. The maximum sequence length is 3,072 tokens.

## E FUTURE DIRECTIONS

Our work introduces a pointer-based command sequence representation and a corresponding model architecture that closely mimics the "select-then-operate" workflow of modern CAD systems. This design is inherently general and not restricted to the operations demonstrated in this work. In principle, it can be extended to a wide range of CAD modeling operations, including *revolve*, *sweep*, *loft*, and *shell*, since all of them require selecting geometric entities and specifying parameters. However, due to the limited accessibility to CAD models involving these operations, they are not included in this study. A key avenue for future research is to expand the dataset to include a wider variety of operations, thereby validating and unlocking the full potential of our approach on a more comprehensive set of modeling tasks.

## F LLM USAGE

We utilized LLMs, including ChatGPT, GitHub Copilot, to assist in the writing and refinement of this paper. The primary use of these models was to improve grammar, clarity, and readability. All content, including the core ideas, experimental results, and conclusions, was conceived and critically reviewed by the authors. The authors take full responsibility for the final version of the manuscript.

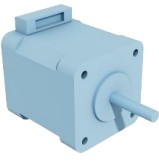 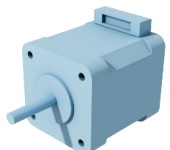 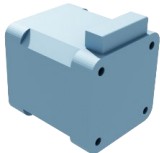 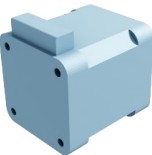

```
You are a senior CAD engineer. I will provide you with four images of
a 3D model. Your task is to:

  1. Generate a one-word name for the object, enclosed in
     <name></name>.
  2. Write a clear and concise one-sentence caption for the object,
     enclosed in <caption></caption>, summarizing its overall shape
     and key structural features. Focus on geometric form, symmetry,
     major extrusions or cutouts, and distinctive elements. Avoid
     interpretation or unnecessary detail.
```

Figure 11: **Prompt for visual description.** This prompt is used with the Qwen2.5-vl-72B model to generate a description of the CAD model's visual appearance.

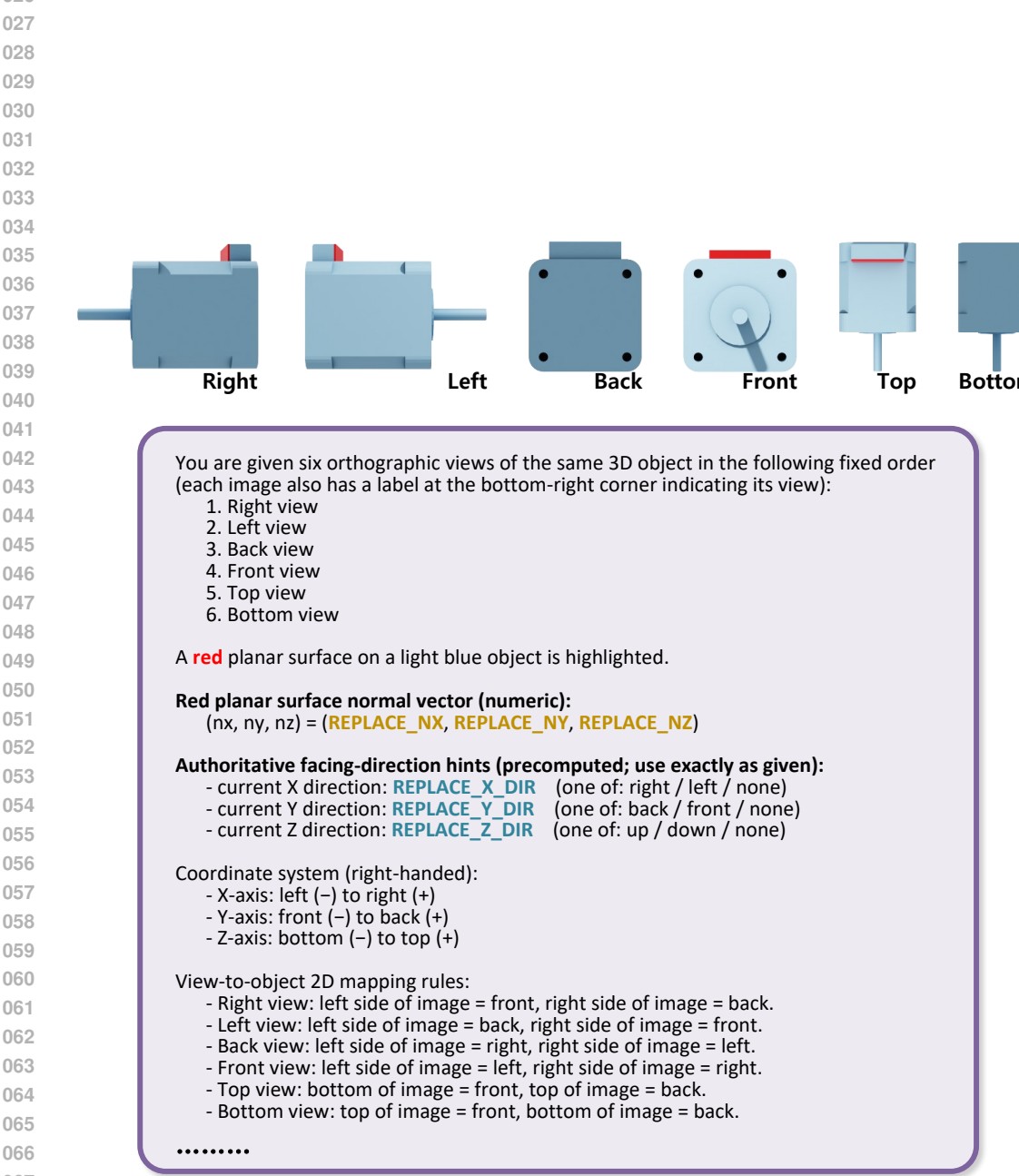

Figure 12: **Prompt for sketch plane description.** This prompt guides the model to describe the relative position of the sketch plane, with placeholders for the normal vector and facing direction being dynamically replaced.

```json
{
  "parts": {
    "part_1": {
      "sketches": {
        "sketch_1": {
          "name": "Cylinder 1",
          "profile_1": {
            "loop_1": {
              "curve_1": {
                "type": "Circle",
                "center": [0.0, 0.0],
                "radius": 0.5
              }
            }
          },
          "coordinate_system": {
            "reference_plane": "Top",
            "rotation": [0.0, 0.0, 0.0],
            "position": [0.0, 0.0, 0.0]
          }
        }
      },
      "extrusion": {
        "name": "Extrude 1",
        "extrude_operation_type": "NewBodyFeatureOperation",
        "extrude_extent_mode": "OneSideFeatureExtentType",
        "extrude_depth_towards_normal": 0.5,
        "extrude_depth_opposite_normal": 0.0
      },
      "description": {
        "label": "Cylinder",
        "caption": "A cylinder with a height equal to
                    half of its diameter.",
        "length": 1.0,
        "width": 1.0,
        "height": 0.5
      }
    },
    "part_2": {
      "sketches": {
        "sketch_1": {
          "name": "Cylinder 2",
          "profile_1": {
            "loop_1": {
              "curve_1": {
                "type": "Circle",
                "center": [0.0, 0.0],
                "radius": 0.25
              }
            }
          },
          "coordinate_system": {
            "description": "On the top face of the cylinder.",
            "reference_plane": "Top",
            "rotation": [0.0, 0.0, 0.0],
            "position": [0.0, 0.0, 0.5]
          }
        }
      },
      "extrusion": {
        "name": "Extrude 2",
        "extrude_operation_type": "NewBodyFeatureOperation",
        "extrude_extent_mode": "OneSideFeatureExtentType",
        "extrude_depth_towards_normal": 0.1,
        "extrude_depth_opposite_normal": 0.0
      },
      "description": {
        "label": "Cylinder",
        "caption": "A very flat cylinder.",
        "length": 0.5,
        "width": 0.5,
        "height": 0.1
      }
    },
    "dimensions": {
      "x_length": 1.0,
      "y_length": 1.0,
      "z_length": 0.6
    }
  }
}
```

```json
{
  "parts": {
    "part_1": {
      "sketches": {
        "sketch_1": {
          "name": "Cylinder 1",
          "profile_1": {
            "loop_1": {
              "curve_1": {
                "type": "Circle",
                "center": [0.0, 0.0],
                "radius": 0.5
              }
            }
          },
          "coordinate_system": {
            "reference_plane": "Top",
            "rotation": [0.0, 0.0, 0.0],
            "position": [0.0, 0.0, 0.0]
          }
        }
      },
      "extrusion": {
        "name": "Extrude 1",
        "extrude_operation_type": "NewBodyFeatureOperation",
        "extrude_extent_mode": "OneSideFeatureExtentType",
        "extrude_depth_towards_normal": 0.5,
        "extrude_depth_opposite_normal": 0.0
      },
      "description": {
        "label": "Cylinder",
        "caption": "A cylinder with a height equal to
                    half of its diameter.",
        "length": 1.0,
        "width": 1.0,
        "height": 0.5
      }
    },
    "part_2": {
      "fillet": {
        "name": "Top Fillet",
        "fillet_edges": {
          "edge_1": {
            "type": "3D Circle",
            "center": [0.0, 0.0, 0.5],
            "via": [1.0, 0.0, 0.5]
          }
        },
        "fillet_radius": 0.1,
        "fillet_tangent_chain": true
      },
      "description": {
        "label": "Cylinder",
        "caption": "A cylinder with a fillet along its top
                    edge."
      }
    },
    "dimensions": {
      "x_length": 1.0,
      "y_length": 1.0,
      "z_length": 0.5
    }
  }
}
```

JSON — Minimal Json

JSON — Minimal Json

Figure 13: **Examples of the minimal JSON structure.** This figure illustrates two structured 'minimal JSONs' format, which integrates visual annotations and key modeling parameters for the language model.

1134
1135
1136

**Annotation Prompt**

**You are a senior CAD engineer. Your task is to read a CAD model construction process described in JSON format and generate clear, natural language instructions that a junior designer can follow to build the model step by step.**
Each part in the JSON is constructed independently and may include:
1. One or more 2D sketches. Each sketch may contain lines, arcs, and circles: (i) A line is defined by a start point and an end point; (ii) An arc is defined by a center point, a start point, a sweep angle, and a direction; (iii) A circle is defined by a center point and a radius.
2. A coordinate system that positions the sketch in 3D space using: (i) A sketch plane (Top, Right, or Front), which defines the basis for the coordinate system; (ii) Rotation angles following Z-Y-X order (first rotate around Z, then Y, then X); (iii) Translation along the x, y, and z directions; (iv) Optionally, a description field may be present, giving a high-level spatial context of the coordinate system's placement in 3D space.
3. An extrusion operation applied to the sketch or sketches. It includes: (i) extrude_operation_type, which defines how the extrusion modifies geometry. This may involve adding material, cutting, intersecting, or creating new bodies or components; (ii) extrude_extent_mode, which defines how far and in which direction the sketch is extruded. Interpret and explain this mode naturally based on the data.
4. A fillet operation, defined by: (i) fillet_radius, which specifies the radius of the fillet; (ii) fillet_tangent_chain, which indicates whether the fillet continues smoothly along tangent edges; (iii) fillet_edges, which specifies the edges to which the fillet is applied.
5. A chamfer operation, defined by: (i) chamfer_distance, which specifies the distance of the chamfer; (ii) chamfer_tangent_chain, which indicates whether the chamfer continues smoothly along tangent edges; (iii) chamfer_edges, which specifies the edges to which the chamfer is applied.
Instructions must follow these rules: 1. Explicitly mention each sketch, even if it has only one shape; 2. Omit unimportant fields like sketch name or body/component name; 3. Wrap all numeric values (coordinates, radii, lengths, distances) in <v></v>; 4. Do not wrap angles, including rotation and sweep_angle; 5. Do not add units; <v></v> implies units; 6. Ignore rotations/translations that are all zero and omit zero-valued axes; 7. Use concise, natural engineering language; 8. You may use paragraphs for readability but avoid lists or step numbers; 9. Ignore captions/visualizations that conflict with construction steps; 10. If a coordinate system has a description, integrate it naturally for high-level context.
You will receive only the JSON content in the prompt. Interpret and process it directly.

Figure 14: Prompt for generating the final natural language description.

1187