# OpenReview forum: "Pointer-CAD: Unifying B-Rep and Command Sequences via Pointer-based Edges & Faces Selection"
_ICLR.cc/2026/Conference — ICLR 2026 Conference Withdrawn Submission_

### Official Review · Reviewer_cRCz · 2025-10-21

**Soundness:** 3
**Presentation:** 3
**Contribution:** 2
**Rating:** 2
**Confidence:** 5

**Summary:**

The paper present Pointer-CAD - a novel method for text-conditioned CAD generation. This is and LLM-based method working in multiple steps, and on each step it consider the B-Rep generated from previous steps. To make this work, authors also present a pointer representation to select faces or edges, enabling fillet or chamfer operations. Pointer-CAD is claimed to outperform DeepCAD, Text2CAD, and CADMium of novel Recap-DeepCAD, and Recap-OmniCAD datasets.

**Strengths:**

- The proposed pointer representation is important to solve existing problem of supporting chamfer and fillet CAD operations, by adding edge and face tokens.
- Authors re-annotate DeepCAD and Omni-CAD datasets resulting in 500k+ novel CAD descriptions.
- Pointer-CAD is claimed to outperform 3 baselines, on 2 datasets, at 13 different metrics.
- The figures of method and CAD predictions clearly describes the proposed solution.

**Weaknesses:**

- No comparison with previous methods on existing benchmarks.
  - As the paper doesn't contribute a novel problem formulation (but novel method and representation), the proposed solution should be compared to existing approaches on existing benchmarks. Conditional CAD generation is a well defined problem with dozens of methods on 3 main datasets (DeepCAD, Fusion360, and CC3D), and 3 main input modalities: point cloud (CAD-SIGNet, CAD-Recode, CAD-MLLM), images (Img2CAD, CADCrafter, CAD-GPT), and text (Text2CAD, Text-to-CadQuery, CAD-Coder, Cadrille). Instead of this authors come with their own benchmarks, namely Recap-DeepCAD and Recap-OmniCAD. This comparison is not enough for ICLR conference.
  - The proposed pointer representation and autoregressive method are orthogonal to input modality, so the method is not limited to text inputs, and can be easily applied to point cloud or image modalities. Namely recent Cadrille and CAD-MLLM (cited in the paper) evaluate their approaches on all 3 modalities simultaneously. I think, to prove the importance of novel representation it is important to demonstrate such a comparisons.

- Misleading and low metrics in main table.
  - The baseline methods CADMium, DeepCAD, Text2CAD on both datasets are reported to achieve 40-50% of invalidity ratio (IR), which basically means that half of predictions are invalid, and the methods don't work. Recent text-to-cad approaches report order of magnitude better IR, namely 1% for Text2CAD, Cadrille, or Text-to-CadQuery. The proposed Pointer-CAD is also claimed to have up to 25% IR, basically not working either, on such simple datasets as DeepCAD.
  - Authors report 3.38 median CD for Text2CAD method on ~DeepCAD dataset, however original Text2CAD reports as low as 0.37 mediand CD on the same data with different annotations. This sounds suspicious to me. Also proposed Pointer-CAD achieves 0.30 median CD on this data, with is I believe far from recent Text-to-CadQuery and Cadrille, reporting 0.22 median CD.
  - All baseline methods are reported without chamfer F1 or fillet F1. I think before adding this to operations to novel Pointer-CAD it is important to show the results of adding them to previous methods. It should be straight forward for methods producing CadQuery code, because CadQuery supports these operations.

- L.015: *"... LLM-based CAD generation by representing CAD as command sequences. But these methods struggle in practical scenarios because command sequence representation does not support entity selection (e.g. faces or edges) ..."* Actually existing LLMs like GPT5 very much support entity selection, chamfer, fillet and all other operations. This can be easily checked by prompting text- or image-conditioned CAD generation in CadQuery format. Also I think even smaller models make step towards this direction, like Text-to-CadQuery mentions face selection (last figure of their supplementary), or Cadrille (and CAD-Recode) first defines some planes and later selects them multiple times.

**Questions:**

- Can the proposed Pointer-CAD be compared with recent text-to-cad approaches Text2CAD, Text-to-CadQuery, and Cadrille on existing Text2CAD benchmark?

- Is the proposed pointer representation useful for other input modalities like image or point cloud?

- Why all methods in Tab. 2 are reported with invalid rate of 25-50%, while recent Text2CAD, Cadrille, and Text-to-Cadquery reports 1%?

- What happens if adding chamfer or fillet operations from your novel dataset to existing methods?

- How is inference time of proposed iterative Pointer-CAD compared to straight forward Text2CAD, Cadrille, Text-to-Cadquery?

---

### Official Review · Reviewer_CKBy · 2025-10-27

**Soundness:** 2
**Presentation:** 2
**Contribution:** 2
**Rating:** 4
**Confidence:** 4

**Summary:**

This work focuses on the task of text-to-CAD generation.
It aims to solve two problems: 1) the existing command sequence representation does not support entity selection and complex editing operations such as chamfer and fillet; and 2) the discretization of continuous variables may result in topological errors.
To address these problems, it proposes Pointer-CAD, which is a LLM-based model that introduces a pointer-based command sequence representation to explicitly incorporate the information of B-Rep representation into sequential modeling.

**Strengths:**

- The challenges proposed in this work, including the lack of support for chamfer or fillet and the lossy discretization of continuous variables, are practical, interesting and valuable.
- The idea of introducing pointer to LLM-based CAD generation has not been explored in previous studies.

**Weaknesses:**

- Are the outputs of graph processing (i.e, the edge and face embedding in lines 268-269) continuous or discretized? Given that LLMs accept discretized tokens as inputs, how are they inputted into LLMs, e.g., directly inputted to LLMs or with some techniques like adapter or other techniques?
- As mentioned in lines 274-276, there are two fully connected layers, one for label token and value token while the other for pointer. Does Fig 2 illustrate all of these two layers and their outputs or only shows the layer and output for label and value token?
- Lines 287-289 mentioned that the LLM's output are matched to the candidate geometric element with highest cosine similarity. What is the goal of this operation, e.g., selecting the target face or edge as mentioned in lines 277-279?
- Lines 279-281 mentioned that there may be more than one element that are valid. How are the number of selected elements decided during the inference?
- The comparison and discussion to CADFusion [1] should be included.

[1] https://arxiv.org/abs/2501.19054 (ICML 2025)

**Questions:**

See weaknesses.

---

### Official Review · Reviewer_opNq · 2025-10-30

**Soundness:** 3
**Presentation:** 3
**Contribution:** 3
**Rating:** 6
**Confidence:** 4

**Summary:**

This paper introduces Pointer-CAD, which addresses limitations of prior command sequence approaches, such as entity selection and quantization-induced topological errors. The authors also develop a large-scale dataset with expert-level annotations. Experiments show that Pointer-CAD significantly reduces segmentation errors and mitigates quantization-induced topological issues.

**Strengths:**

1. The motivation to support complex editing operations, such as chamfering or filleting, is well justified.
2. The proposed method of unifying B-rep and command sequences is novel and conceptually sound.
3. Based on the qualitative experimental results, the proposed approach appears to be effective.

**Weaknesses:**

1. The paper lacks a sensitivity analysis of hyperparameters, such as λv and λp.
2. The implementation of Pointer-CAD only uses Qwen2.5-0.5B as the base LLM. It would be interesting to investigate whether performance improves significantly with larger-scale LLMs.
3. For the captioned 575,559 samples, there seems to be no filtering mechanism to handle lower-quality data.

**Questions:**

Please address the weaknesses above.

---

### Official Review · Reviewer_qjCD · 2025-11-01

**Soundness:** 3
**Presentation:** 3
**Contribution:** 2
**Rating:** 2
**Confidence:** 4

**Summary:**

The core objective of this paper is to address two key limitations of existing large language model (LLM)-based CAD generation methods: "inability to support complex entity selection operations" and "topological errors caused by the discretization of continuous variables."The paper proposes a pointer-based command sequence representation and the corresponding "Pointer-CAD" framework: Leveraging a "Pointer" mechanism to directly reference geometric entities (e.g., edges and faces) in the boundary representation (B-rep), it decomposes CAD modeling into three ordered steps: "sketch-extrusion," "chamfering," and "filleting." Each step generation relies on both the input text description and the preceding B-rep model. Meanwhile, a multimodal fusion module is designed to integrate textual semantics and B-rep geometric features. The prediction of pointers and tokens is optimized through LLM fine-tuning and a contrastive loss function, and a Recap-OmniCAD+ dataset containing 575,000 models is constructed to support training. Experiments demonstrate that Pointer-CAD can effectively generate complex geometric structures, reducing the segmentation error (SegE) to the order of \(10^{-3}\) (a 100-fold improvement over traditional methods), and it is the only text-driven CAD generation method that supports chamfering and filleting operations.

**Strengths:**

Breaking through the entity selection bottleneck to support industrial-grade complex operationsExisting LLM-based CAD generation methods (e.g., Text2CAD, CADmium) only include numerical parameters in their command sequences, making it impossible to implement editing operations that rely on edge/face selection, such as "chamfering" and "filleting."
Eliminating quantization errors and improving topological and geometric accuracyTraditional methods need to discretize continuous variables (e.g., coordinates, extrusion distances) into limited quantization levels, which easily leads to topological issues such as "new curves failing to align with existing edges" and "sketch plane misalignment." Pointer-CAD fundamentally addresses this problem through the pointer mechanism: Sketch planes directly select faces in the B-rep via "face pointers," avoiding 3D parameter regression errors; curves align with existing entities through "edge pointers," eliminating the need for discrete coordinate-based positioning.
Constructing high-quality datasetsTo support the learning of the pointer mechanism and complex operations, the paper designs an automated annotation pipeline based on Qwen2.5-VL/Qwen2.5: It renders multi-view images of CAD models to generate geometric descriptions and spatial location texts; converts raw data into "concise JSON" with embedded text annotations; and ultimately constructs the Recap-OmniCAD+ dataset (575,000 models) containing chamfering/filleting operations and the comparative Recap-DeepCAD dataset (176,000 models). The datasets cover complex topologies (holes, multi-body components), are more aligned with industrial scenarios, and provide sufficient samples for model generalization.

**Weaknesses:**

Chamfer operations adopt a sequence representation, without exploring the effectiveness of code generation modesThe paper encodes operations such as chamfering and filleting into a sequence representation of "pointer + token," relying on LLMs to generate sequences that are then converted into B-rep operations. However, mature parametric programming tools (e.g., CADQuery) already exist in the CAD domain, which can directly define chamfer operations through code. Such code generation modes may be more intuitive and eliminate the need for additional "vector conversion modules" to parse sequences into operations. The paper fails to compare the efficiency differences between "sequence representation" and "code generation": for instance, whether code generation can reduce "sequence parsing errors," whether it is more easily adaptable to the native APIs of CAD software, and whether it is more efficient for complex models (with multiple chamfer edges or multi-parameter combinations). These issues remain unexplored.
Limited extension of operation types, failing to cover complex scenarios such as "subpart assembly"The paper only extends CAD operations to three categories: "sketch-extrusion," "chamfering," and "filleting," without involving more complex operation types in industrial modeling, especially "subpart assembly"—for example, positioning and assembling another predefined subpart on a specified surface (e.g., installing a bolt model on the top surface of a cube). Such operations not only require "selecting the target surface" but also involve handling more complex geometric relationships such as "coordinate alignment between subparts and parent models" and "assembly constraints (e.g., fitting, coaxiality)." The pointer mechanism of the existing framework only supports "edge/face selection" and cannot express assembly constraints; furthermore, the dataset does not include subpart annotations, and the model is not designed with generation logic for "subpart calling and positioning," resulting in functionality limited to "single part generation," which 难以满足 the assembly design requirements of actual products.
Dependence on B-rep historical information may lead to accumulation of multi-step errorsEach step generation in Pointer-CAD relies on the geometric information of the preceding B-rep. If a minor error occurs in a certain step (e.g., pointer selection deviation, inaccurate numerical parameters), subsequent steps will continue generating based on the erroneous B-rep, leading to error accumulation. For example: a deviation in the prediction of numerical tokens during the first extrusion step results in a smaller cube height; the second step selects the "top surface" of this cube for chamfering, which may fail due to insufficient top surface area. The paper does not evaluate the relationship between "number of steps and error accumulation" nor design an "error correction mechanism" (e.g., backtracking to adjust previous steps), which may pose a risk of precision degradation in multi-step complex models (with 10+ steps).

**Questions:**

See Weaknesses

---

### Author Response · Authors · 2025-11-14

We are grateful for the reviewers’ time and constructive feedback, which we will carefully incorporate to refine and improve our paper.

---

### Note · Authors · 2025-11-14

I have read and agree with the venue's withdrawal policy on behalf of myself and my co-authors.